# Quality of Life Following Pelvic Organ Prolapse Treatments in Women: A Systematic Review and Meta-Analysis

**DOI:** 10.3390/jcm11237166

**Published:** 2022-12-01

**Authors:** Zinat Ghanbari, Marjan Ghaemi, Arman Shafiee, Parivash Jelodarian, Reihaneh Sadat Hosseini, Shahla Pouyamoghaddam, Ali Montazeri

**Affiliations:** 1Vali-E-Asr Reproductive Health Research Center, Family Health Research Institute, Tehran University of Medical Sciences, Tehran 1417613151, Iran; 2Student Research Committee, School of Medicine, Alborz University of Medical Sciences, Karaj 3149969415, Iran; 3Department of Obstetrics and Gynecology, Fertility Infertility and Perinatology Research Center, School of Medicine, Ahvaz Jundishapur University of Medical Sciences, Ahvaz 6135715794, Iran; 4Health Metric Research Center, Iranian Institute for Health Sciences Research, ACECR, Tehran 1983963113, Iran

**Keywords:** pelvic organ prolapse, quality of life, systematic review

## Abstract

Introduction: Quality of life (QoL) improvement is one of the main outcomes in the management of pelvic organ prolapse as a chronic illness in women. This systematic review aimed to investigate the impact of surgical or pessary treatment for pelvic organ prolapse (POP) on quality of life. Methods: Preferred Reporting Items for Systematic Reviews and Meta-Analyses (PRISMA) was applied. Electronic databases, including PubMed, Scopus, and Web of Science, were searched for original articles that evaluated the QoL before and after surgical interventions or pessary in pelvic organ prolapse from 1 January 2012 until 30 June 2022 with a combination of proper keywords. Included studies were categorized based on interventions, and they were tabulated to summarize the results. Results: Overall, 587 citations were retrieved. Of these, 76 articles were found eligible for final review. Overall, three categories of intervention were identified: vaginal surgeries (47 studies), abdominal surgeries (18 studies), and pessary intervention (11 studies). Almost all interventions were associated with improved quality of life. The results of the meta-analysis showed a significant association between the employment of surgical approach techniques (including vaginal and abdominal surgeries) and the quality of life (Pelvic Floor Distress Inventory (PFDI) (MD: −48.08, 95% CI: −62.34 to −33.77, *p*-value < 0.01), Pelvic Floor Impact Questionnaire (PFIQ) (MD: −33.41, 95% CI: −43.48 to −23.34, *p* < 0.01)) and sexual activity of patients with pelvic organ prolapse (Pelvic Organ Prolapse/Urinary Incontinence Sexual Function Questionnaire (PISQ) (MD: 4.84, 95% CI: 1.75 to 7.92, *p* < 0.01)). Furthermore, narrative synthesis for studies investigating the effect of the pessary approach showed a positive association between the use of this instrument and improvement in the quality of life and sexual activity. Conclusions: The results of our study revealed a significant improvement in the women’s quality of life following abdominal and vaginal reconstructive surgery. The use of pessary was also associated with increased patient quality of life.

## 1. Introduction

Pelvic organ prolapse (POP) occurs due to weakness of the supportive tissues of the pelvic organs, which may lead to prolapse of the anterior and/or posterior vaginal wall, the uterus (cervix), or the apex of the vagina (vaginal vault or cuff scar after hysterectomy) [1]. The prevalence of POP is currently increasing due to extended life expectancies and childbearing in low-resource areas [2]. Pelvic prolapses are not always symptomatic and can lead to discomfort in the vagina and changes in bladder and bowel function that can greatly affect women’s quality of life [3], with general, social, psychological, and sexual impacts [4]. Therefore, improving the quality of life is one of the main outcomes in the management of pelvic organ prolapse in women [5].

Surgical interventions for POP include repairing with native tissue or mesh and minimally invasive surgeries such as laparoscopic or robotic techniques, which are increasing in popularity [6]. The selection of the intervention depends on several factors, such as the site and severity of the POP; additional symptoms that affect urinary, bowel, or sexual function; the wish to preserve the uterus; and the surgeon’s choice and ability. Surgical treatment options include vaginal or abdominal (laparotomy, laparoscopy, and, more recently, robotic approach) [7]. There are also conservative interventions, which are defined as non-surgical methods such as optimizing lifestyle (weight loss and avoiding heavy lifting or coughing) and physical therapies [8]. In the last decades, pessaries, which have existed since the beginning of recorded history, have also been used in women with POP [9]. These are removable devices that provide support after prolapse [9]. Various instruments have been designed to assess the quality of life (QoL). Some of them evaluate the general aspect, whereas others, such as the Pelvic Organ Prolapse Distress Inventory (POPDI-6) and Pelvic Organ Prolapse Impact Questionnaire (POPIQ-7), are specific for POP [10,11]. Some questionnaires are dedicated to the quality of sex life [12].

QoL studies for POP are very diverse, with different methods, instruments, and follow-ups. Therefore, studies that summarize the results and provide final recommendations are scarce. Performing a systematic review is the best way to summarize the effects of POP treatment on QoL. A similar study was performed in 2012 by Doaee et al. that examined articles over the past ten years [13]. Due to the advances in urogynecological surgery and other interventional methods such as pessaries, updating this data seems necessary. The current study aims to review those studies that have focused on changing the QoL by means of surgery or pessary for POP management.

## 2. Methods

The current systematic review was designed to review QoL in women before and after surgery or pessary intervention for POP management in English biomedical journals. The study is reported based on the Preferred Reporting Items for Systematic reviews and Meta-Analyses (PRISMA) statement [14].

### 2.1. Eligibility Criteria

The inclusion criteria were as follows: (1) study design: all original articles including randomized clinical trials, observational studies (cross-sectional, case-control, or cohort), and editorials/letters; (2) patient population and intervention: adult women with POP who received surgical treatments or pessary intervention for POP management; and (3) outcome: evaluated quality of life using available questionnaires. Review articles, opinions or guidelines, conference abstracts, non-peer-reviewed papers, case reports, unpublished reports, and articles in which the date and location of the study were not specified were excluded.

### 2.2. Information Sources

The initial search was undertaken in three main databases including articles published in MEDLINE (through PubMed), Scopus, and Web of Science from 1 January 2012 until 31 June 2022. Additionally, a manual search in the reference section of the relevant studies was done to obtain possible publications that were missed in our electronic search.

### 2.3. Search Strategy

To retrieve citations on the topic based on the medical subject heading (Mesh), a combination of the following keywords was used: ‘pelvic organ prolapses’, ‘quality of life’, and ‘treatment’. The study time frame was also applied to all databases. A full search strategy for each database is available in the Appendix A.

### 2.4. Study Selection

Titles and abstracts were independently reviewed for eligibility by three authors (PJ, RH, and SP), and non-relevant or duplicate studies that did not meet the inclusion criteria were excluded. In cases of disagreement, the problem was resolved by discussion and the main author (ZG.). After initial screening, the full texts of the articles were reviewed, and the unrelated articles were removed.

### 2.5. Quality Assessment

NS and AS independently assessed the quality of the included studies using the National Heart, Lung, and Blood Institute (NHLBI) tools [15]. Based on the design of a study (whether a randomized controlled trial or a cohort), an individual checklist that contains 14 signaling questions for assessing the quality of each study was used. Briefly, studies scoring nine or more “Yes” answers were marked as “Good”, studies scoring between seven or eight were marked as “Fair”, and studies rating less than seven were marked as “Poor” quality.

### 2.6. Data Extraction

The data—including the first author, publication date and study design, intervention mode, using mesh or native tissue, sample size, stage of prolapse and prolapse type, main findings, and the instrument used to measure the QoL, as well as the follow-up duration and quality of papers—were extracted and tabulated.

### 2.7. Statistical Analysis

We used the mean difference (MD) of the total quality of life questionnaire values before and after the intervention. Using a random effect model, the quantitative values of each study were pooled separately. If the MD was not given in the specific study, an estimate was made using Excel calculators [16]. The Pelvic Floor Distress Inventory Questionnaire (PFDI) and Pelvic Floor Impact Questionnaire (PFIQ-7) were used as the main measures for pooling the results of the studies regarding the QoL of patients. Furthermore, the PISQ and FSFI questionnaires were used to pool the data regarding the sexual activity of patients after the intervention. We used Cochran’s Q statistic (Q-test) and the I2 to assess heterogeneity. I2 value >75% indicated a high amount of heterogeneity. Publication bias was assessed by visual inspection of the funnel plot, and Egger’s test with a significance level of 0.05 was used to evaluate the publication bias. All analyses were statistically significant with a *p*-value < 0.05. The analyses were performed using R-4.1.3 software and the Meta package (R Core Team, Vienna, Austria; available at https://www.R-project.org/, accessed on 6 January 2022).

### 2.8. Registration Statement

We have written the protocol of this study, but due to the high diversity of studies and quality of life assessment methods, we did not register it due to the high possibility of changes in the protocol

## 3. Results

### 3.1. Statistics

Overall, 587 citations were found to be eligible. After excluding 435 duplicates, 122 studies were screened by the title and abstract. By excluding 21 citations, 100 full text articles were reviewed, and 76 articles were finally eligible for final evaluations. The flowchart of the study is depicted in Figure 1.

### 3.2. Instrument Used

A wide variety of questionnaires were used to measure patient-reported outcomes, including QoL measures. Of these, some were the general measures, and several instruments were pelvic-specific QoL questionnaires. Some instruments are used for measuring sexual QoL in sexually active women. The most common general instruments were the 36-Item Short Form Survey (SF-36) and Patient Global Impression of Improvement (PGI-I), while the pelvic-specific measures were the Pelvic Floor Distress Inventory Questionnaire-20 (PFDI- 20), Pelvic Organ Prolapse/Urinary Incontinence Sexual Questionnaire-12 (PISQ-12), Prolapse Quality of Life (P-QOL), and Pelvic Floor Impact Questionnaire-7 (PFIQ-7). The Sexual QoL was measured by the Female Sexual Function Index (FSFI). The questionnaires are listed in Table 1.

### 3.3. General Findings

To facilitate reporting the results, papers were classified into three categories: vaginal surgeries (47 articles), abdominal surgeries (18 articles), and pessary interventions (11 articles). These are categorized in Figure 2.

#### 3.3.1. Vaginal Surgeries

There were two main procedures for vaginal surgeries: reconstructive (in two subgroups, including repair with natural tissue [17,19,20,23,24,31,36,37,38,60,61,63,64,65,66] (Table 2) and repair with mesh [7,9,12,21,25,26,29,32,35,39,40,41,42,43,44,55,62,67,68,69,78,82,83,85]) (Table 3) and obliterative surgeries [18,27,60,71,72,73,86] (Table 4). In one study, obliterative surgery and sacrospinous fixation in older postmenopausal women were compared, and the QoL was better in the sacrospinous group [18]. These results are the opposite of another article with the same method and population that showed that obliterative surgery versus reconstructive acted better in improving the QoL [28,72,79].

Overall, 60.5% (23/38) of the reconstructive studies used mesh for repairing. In a recent study, transvaginal mesh surgery and laparoscopic mesh sacropexy had similar results [9]. In addition, in one cohort study, QoL was measured after POP surgery with or without mesh and did not differentiate between individuals with and without mesh [31]. All studies investigated the QoL in the first surgery, except one, which evaluated the transvaginal bilateral sacrospinous fixation after the second recurrence of vaginal vault prolapse, which improved the QoL and sexual function [61].

#### 3.3.2. Abdominal Surgeries

Overall, eight [33,47,48,56,74,75,76,87] and two studies [53,54] were dedicated to just laparoscopic or robotic approaches, respectively (Table 5). Four citations compared vaginal-assisted laparoscopic sacrohysteropexy and vaginal hysterectomy with vaginal vault suspension for advanced uterine prolapse, which has similar results [45,49,50,55]. Another study compared two methods of laparoscopic and robotic ventral mesh rectopexy, and the results had no difference [51]. One study compared robotic and vaginal sacropexy with comparable results, and in one study, three methods were compared [46].

Only one study was on laparotomy [77], and another citation evaluated the difference between vaginal (using native tissue or with a mesh prolapse) and abdominal (open or robotic abdominal sacrocolpopexy), with results in favor of the abdominal group [78].

#### 3.3.3. Pessary Intervention

Three studies compared pessary and surgery [79,80,81] (Table 6). In one, the women who underwent surgery had better QoL [84], whereas in other studies, the QoL after two interventions had no differences. All studies used ring pessaries [28,77], except three citations that used Gellhorn/cube pessaries [58,59,89].

### 3.4. Overall Findings and Meta-Analysis

Almost all interventions, including surgery and pessary interventions, were associated with improved quality of life. In cases where two different surgical or surgical and pessary methods were compared, the results were inconsistent.

Among the included studies, fifteen studies used the PFDI questionnaire to estimate the QoL [9,31,33,36,37,40,41,42,45,47,51,52,55,56]. The pooled results showed a significant improvement in QoL after surgical interventions (MD: −48.06, 95% CI: −62.34 to −33.77, I2: 97%, *p* < 0.01) (Figure 3). Visual inspection of the funnel plot and results of Egger’s test for funnel plot asymmetry (*p* = 0.17) indicated no possible source of publication bias (Figure 4).

Eleven studies used the PFIQ questionnaire to estimate the QoL [9,36,37,40,42,43,45,51,55,56]. The pooled results showed a significant improvement in QoL after surgical interventions (MD: −33.41, 95% CI: −43.48 to −23.34, I2: 99%, *p* < 0.01) (Figure 5). Visual inspection of the funnel plot and results of Egger’s test for funnel plot asymmetry (*p* = 0.52) indicated no possible source of publication bias (Figure 6).

For estimating the effect of surgical intervention on sexual activity, 14 studies were included [9,12,20,24,37,39,42,43,55,56,61,75,76,85]. Among them, 10 studies used the PISQ questionnaire. The pooled results showed a significant improvement in sexual function after surgical interventions (MD: 4.84, 95% CI: 1.75 to 7.92, I2: 98%, *p* < 0.01) (Figure 7). Visual inspection of the funnel plot and results of Egger’s test for funnel plot asymmetry (*p* = 0.01) indicated a possible source of publication bias (Figure 8).

## 4. Discussion

In this systematic review, we found that the QoL was significantly improved in women after surgical or pessary interventions for the management of POP. We performed a meta-analysis of QoL and sexual activity questionnaires. The results of the meta-analysis showed a significant association between surgical approach techniques (including vaginal and abdominal surgeries) and improved QoL and sexual activity of patients with POP. Due to vast heterogeneity in the pessary approach and QoL questionnaires used, we were not able to pool the results of studies regarding pessary. However, descriptive analysis showed an improved quality of life in these patients.

Interventional and observational studies dedicated to POP surgeries or using pessaries have grown significantly in the last decade [90,91]. On the other hand, with the emergence of specialized QoL questionnaires for pelvic organ prolapse and its symptoms, we encounter a significant amount of data concerning QoL in different methods [92].

As mentioned before, we divided the studies into three parts based on the intervention approach. In vaginal studies, we also dealt with reconstructive and obliterative methods. Increasing numbers of elderly women and their co-morbidities have increased the preference for obliterative vaginal surgery, due to high levels of durability with lower rates of morbidity. Obliterative methods seem to be a good method for older women who are not sexually active and could not tolerate major surgeries with good durability and relative ease of surgery [90,93]. Few studies evaluated the QoL of patients following obliterative methods. The results of these surgeries were satisfactory, and in two studies that were compared with other methods, the results were contradictory [18,72].

Reconstructive methods using synthetic meshes for pelvic organ prolapse and/or stress urinary incontinence have been popular since the mid-1990s [50]. Mesh repairs were effective as traditional repairs [81] and improve QoL [26]. Patients may benefit from anatomical stability when the risks are justifiable [7], with a low rate of recurrence and few complications [7]. However, reports of mesh-related complications are increasing [7,90], and the Food and Drug Administration (FDA) recently warned about using transvaginal mesh due to adverse events including vaginal erosion, dyspareunia, pain, and infection [57,92]. However, there is no consensus in this regard, and the use of mesh is recommended by some experts.

With regard to abdominal approaches, three different approaches were evaluated in the studies. Laparoscopic sacrocolpopexy and sacrohysteropexy have been demonstrated to be effective and safe with faster recovery time, shorter operating time, lower blood loss, lower scar tissue, lower pain, and minimally invasive nature compared to the abdominal approaches [94]. However, these procedures have been associated with some complications, such as stress urinary incontinence [95], defecation problems [96], and injuries of the presacral venous plexus [97]. On the other hand, robotic surgeries have recently been introduced with good results and improvement in QoL. A systematic review and meta-analysis showed less intraoperative bleeding, lower incidence of postoperative complications, and shorter hospital stay for RVMR compared with LVMR, but found no differences in rates of recurrence, conversion, or reoperation [98].

Finally, the vaginal pessary is a conservative treatment for pelvic organ prolapse and can be offered as the first-line treatment in most patients [99]. Non-surgical modalities such as pessary are the best choice for older women because most of them have some type of cardiovascular disease [100] or diabetes mellitus [101]. Among various vaginal pessaries, the ring pessary is the most common type because it is convenient to insert and remove and has acceptable continuation rate and manageable adverse events. In addition, a ring with support pessary is a safe and effective conservative treatment for POP; it not only relieves bothersome prolapse and urinary symptoms but also significantly decreases their impacts on health-related QoL. One-quarter of the patients discontinued using pessary mainly due to dissatisfaction with pessary effectiveness or adverse events such as vaginal discharge or vaginal erosion [102]. Some women also prefer surgery after a while. Pessary treatment is continued beyond 12 months after initial placement by 63% of patients [34]. Comparative studies between pessary and surgery are not numerous, and their results are contradictory. In one study, women had the same quality of life [80], while in another study, women who underwent surgery reported a better QoL than pessary users [34].

In this study, we also had some limitations. First, due to the differences in the questionnaires used, variation in follow-up duration, and different methods of surgery, there was a significant amount of heterogeneity in the results of our meta-analysis. Furthermore, we only investigate a limited number of questionnaires available for assessing QoL patients with POP in our quantitative synthesis. However, it is worth noting that the results of other questionnaires were in line with our meta-analysis, which showed improved QoL of these patients after the aforementioned approaches were used. The conclusions of the meta-analysis were made mostly from the results of a limited number of observational studies, which lowers the certainty of evidence because of their observational nature. However, the results of almost all studies included in this systematic review were in line with our findings. Pelvic floor surgeries are relatively new and have made great strides in the last decade. It is suggested that, in future studies, the methods used with mesh be analyzed in a more specialized way.

## 5. Conclusions

QoL is significantly improved in women after surgical or pessary interventions for management of POP. Due to the daily progress of urogynecological and modern technologies in surgery and less invasive treatments, long-term cohort studies are recommended to evaluate the QoL in these people.

## Figures and Tables

**Figure 1 jcm-11-07166-f001:**
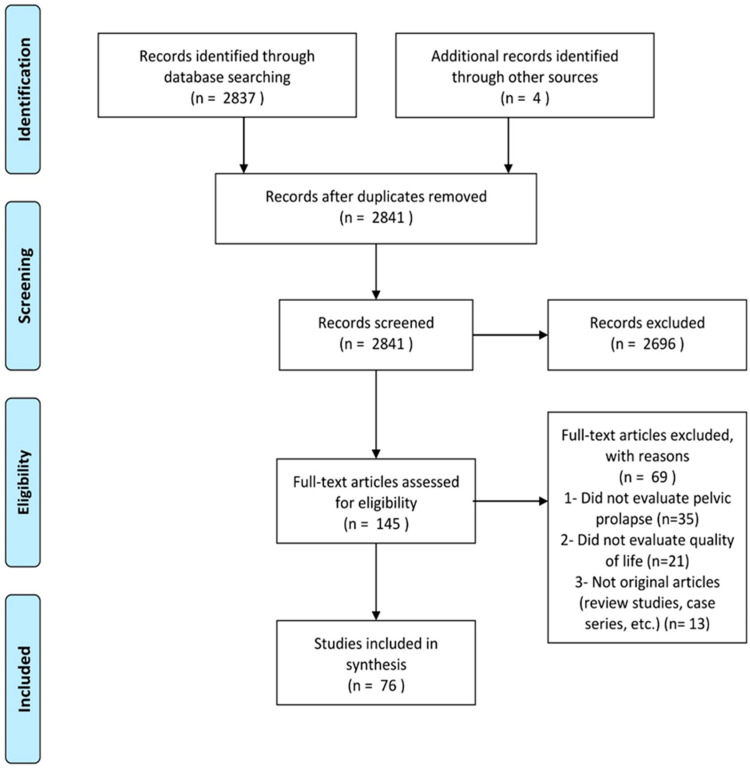
Flow diagram of the recruiting studies according to PRISMA.

**Figure 2 jcm-11-07166-f002:**
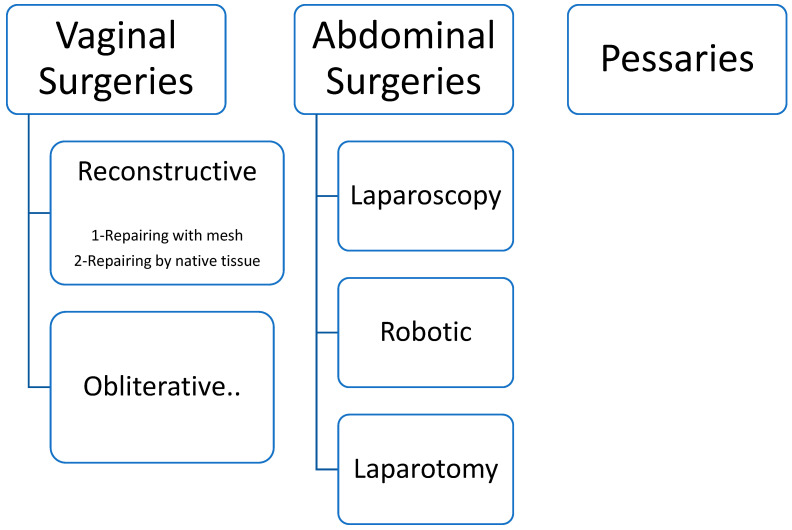
Categorization of the included studies.

**Figure 3 jcm-11-07166-f003:**
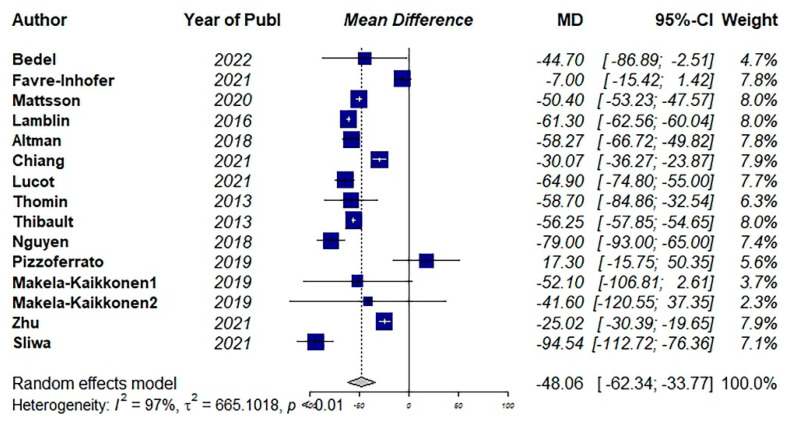
Pooled mean difference of the effect of surgical intervention (vaginal and abdominal surgery) on total quality of life score using the Pelvic Floor Distress Inventory Questionnaire (PFDI) [31,33,36,37,40,41,42,45,47,51,52,55,56].

**Figure 4 jcm-11-07166-f004:**
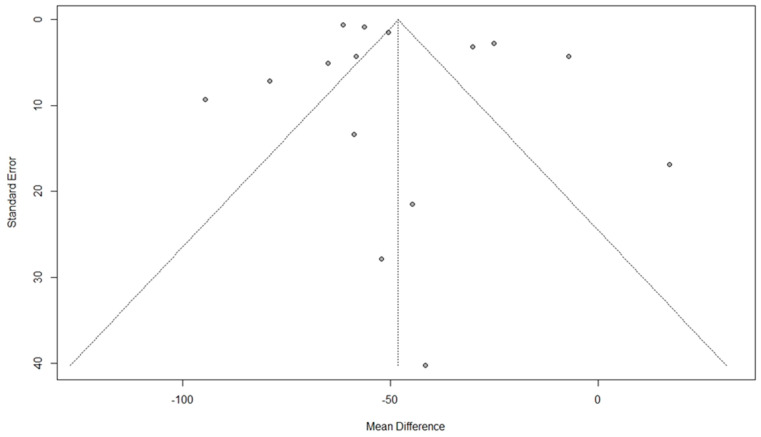
Funnel plot for the results of the effect of surgical intervention (vaginal and abdominal surgery) on total quality of life score using the Pelvic Floor Distress Inventory Questionnaire [PFDI].

**Figure 5 jcm-11-07166-f005:**
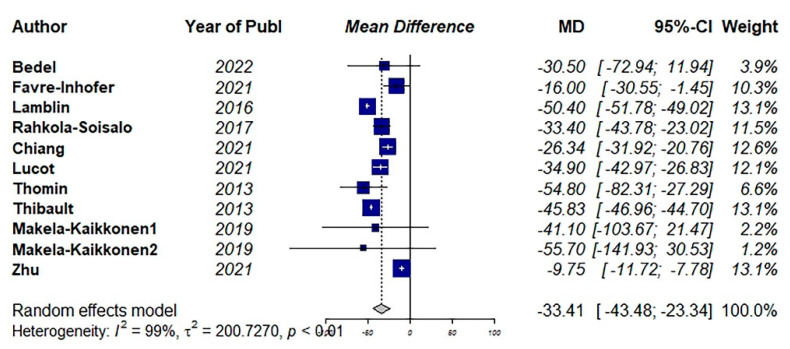
Pooled mean difference of the effect of surgical intervention (vaginal and abdominal surgery) on total quality of life score using Pelvic Floor Impact Questionnaire (PFIQ) [9,36,37,40,42,43,45,51,55].

**Figure 6 jcm-11-07166-f006:**
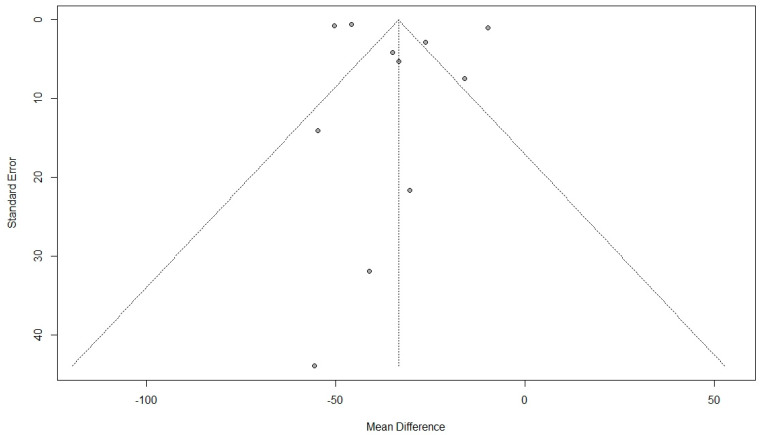
Funnel plot for the results of the effect of surgical intervention (vaginal and abdominal surgery) on total quality of life score using Pelvic Floor Impact Questionnaire (PFIQ).

**Figure 7 jcm-11-07166-f007:**
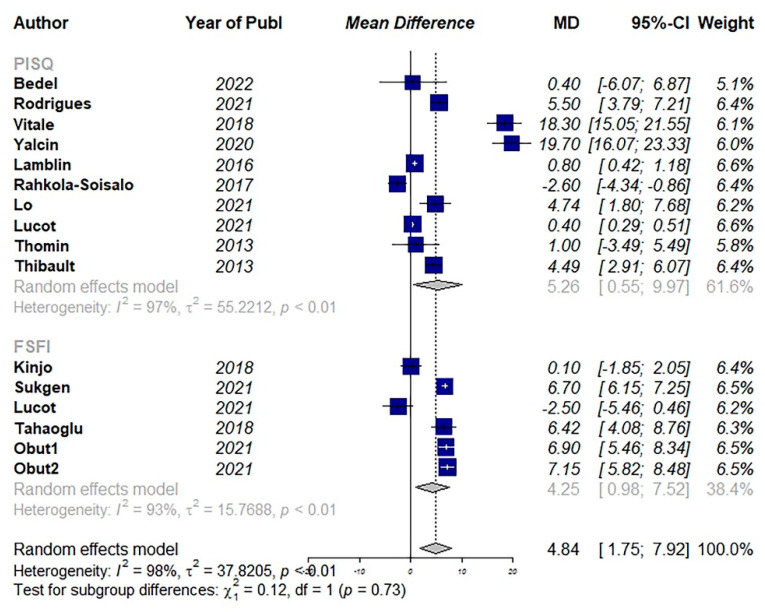
Pooled mean difference of the effect of surgical intervention (vaginal and abdominal surgery) on total quality of sexual activity. (PSIQ: Pelvic Organ Prolapse/Urinary Incontinence Sexual Function Questionnaire, FSFI: Female Sexual Function Index) [9,20,24,37,39,42,43,55,56,61,75,76,85].

**Figure 8 jcm-11-07166-f008:**
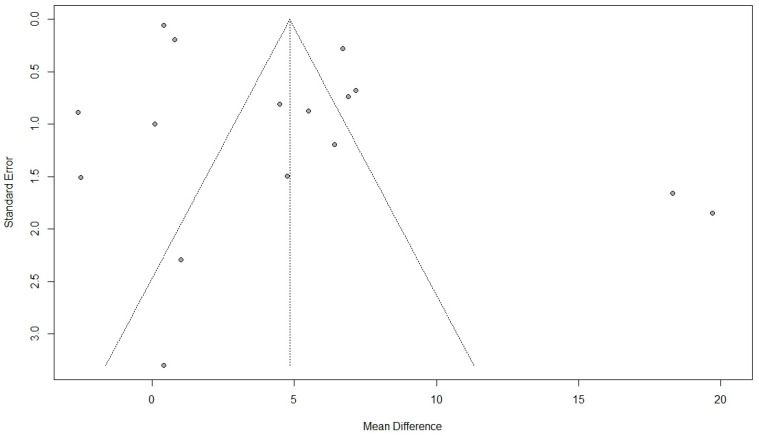
Funnel plot for the results of the effect of surgical intervention (vaginal and abdominal surgery) on total quality of sexual activity.

**Table 1 jcm-11-07166-t001:** Questionnaires used in the recruited studies.

General Instruments		References
	World Health Organization Quality of Life Questionnaire-Brief version (WHOQOL-BREF)	[17,18]
	King’s Health Questionnaire (KHQ)	[19,20,21,22]
	Patient Health Questionnaire-9 (PHQ-9	[23]
	36-Item Short Form Survey (SF-36)	[22,24,25,26,27,28]
	12-Item Short Form Survey (SF-12)	[29,30]
	Patient Global Impression of Improvement (PGI-I)	[23,31,32,33,34]
	European quality of Life Five Dimension (Euro Qol EQ-5D)	[9,21,33]
	Satisfaction With Life Scale (SWLS)	[35]
	Patient Satisfaction Index (PSI)	[21]
**Pelvic-specific instruments**		
	Pelvic Floor Distress Inventory Questionnaire-20 (PFDI- 20) *	[9,27,31,33,36,37,38,39,40,41,42,43,44,45,46,47,48,49,50,51,52,53,54,55,56,57,58,59]
	Pelvic Organ Prolapse/Urinary Incontinence Sexual Questionnaire-12 (PISQ-12)	[9,20,24,33,37,42,43,44,48,50,55,56,57,60,61,62]
	Prolapse Quality of Life (P-QOL)	[7,12,22,23,32,35,47,48,60,63,64,65,66,67,68,69,70,71,72,73,74,75,76,77,78,79]
	Pelvic Organ Prolapse Symptom Score (POP-SS)	[23]
	Body Image in the Pelvic Organ Prolapse (BIPOP)	[23]
	International Consultation on Incontinence Questionnaire-Vaginal Symptoms (ICIQ-VS)	[28,30,32,49,66,80]
	Pelvic Floor Impact Questionnaire-7 (PFIQ-7)	[9,27,33,36,37,39,40,42,43,45,46,51,54,55,56,57,58,59,81]
	International Consultation on Incontinence Questionnaire-Urinary Incontinence Short Form (ICIQ-UI SF)	[9,33,57,80]
	International Consultation on Incontinence Questionnaire-Quality of Life	[49]
	Sheffield Prolapse Symptoms Questionnaire (SPSQ)	[21]
	Patient Assessment of Constipation Quality of Life Questionnaire (Pac-QOL)	[9]
	German Pelvic Organ Prolapse Questionnaire	[82,83,84]
**Sexual functioning instruments**		
	Female Sexual Function Index (FSFI)	[9,13,26,75,76]

* The PFDI-20 consists of three parts, including the Pelvic Organ Prolapse Distress Inventory (POPDI-6), the Urinary Distress Inventory (UDI-6), and the Colo-Rectal-Anal Distress Inventory (CRADI-8).

**Table 2 jcm-11-07166-t002:** The characteristics of the studies on the effectiveness of reconstructive vaginal surgeries by native tissue on the quality of life in women with pelvic organ prolapse.

First Author [Ref]	Year/Study Design	Sample Size	Questionnaire	Prolapse Type and Stage	Intervention	Follow Up	Main Findings	Quality
Dhital [17]	2013/Longitudinal study	252	WHOQOL-BREF	All types ≥ 3	Vaginal hysterectomy with pelvic floor surgery	3 months	At three months post-surgery: (1) Physical domain improved from 11.2 to 13.5; (2) Psychological domain improved from 11.6 to 13.8; (3) Social relationships domain improved from 13.6 to 15; (4) Environmental domain improved from 12.9 to 14. All changes were significant.	Good
De Oliveria [66]	2014/Observational cohort	65	P-QOL + ICIQ-VS	All types NR	Anterior repair, posterior repair, sacrocolpopexy, total vaginal hysterectomy	6 months	Preoperative scores of all domains on the ICIQ-VS and P-QoL were significantly higher than those at three and six months after surgery (*p* < 0.0001).	Fair
Klapdor [65]	2017/Prospective observational study	170	P-QOL	Apical NR	Sacrocolpopexy hysterectomy, anterior colporrhaphy, posterior colporrhaphy, vaginosacrocolporectopexy, lateral repair	26.5 months	The P-QOL scores were either low (<40) or very low (<20) after the intervention, indicating a high QOL among patients.	Good
Jelovsek [38]	2018/Randomized clinical Trial	374	PFDI	Apical ≥ 2	Uterosacral ligament suspension vs. sacrospinous ligament fixation	6–60 months	Improvements in Pelvic Organ Prolapse Distress Inventory scores were −59.4 in the BPMT group and −61.8 in the usual care group.	Good
Vitale [61]	2018/Prospective observational study	20	PISQ-12, SF-36	Apical 2	Transvaginal bilateral sacrospinous fixation after second relapse	12 months	A significant improvement in SF-36 and PISQ-12 scores were found.	Fair
Yalcin [24]	2020/Prospective observational study	26	SF-36 + PISQ-12	Apical ≥ 2	Sacrospinous ligament fixation with vaginal hysterectomy	6 months	QoL increased compared to baseline values for all the categories including physical functioning, bodily pain, physical health, general health, vitality, social activity, emotional state, and mental health (*p* < 0.001)	Good
Mattsson [31]	2020/Prospective observational study	3515	PFDI-20 + PGI	All types ≥ 2	Sacrocolpopexy or vaginal mesh laparoscopy	24 months	72% of the participants reported a clinically significant improvement in Pelvic Floor Distress Inventory-20 at the 2-year follow-up.	Good
Rechberger [64]	2020/ Prospective observational study	200	ICIQ-SF	NR ≥ 2	Vaginal native tissue repair	12 months	Significant reduction in ICIQ-SF results after surgery.	Fair
Karaca [60]	2021/Retrospective cohort	118	P-QOL + PISQ-12	All types ≥ 3	McCall culdoplasty vs. sacrospinous ligament fixation	12 months	McCall culdoplasty acted better (21.4(10.1) vs. 30.8(15.2); *p* = 0.03)	Good
Derpapas [63]	2021/ Pilot randomized controlled trial	22	P-QOL	Posterior Any stages	Posterior colporrhaphy vs. vaginal plication of the posterior vaginal wall	6 months	Significant QoL improvement in both groups after surgery. No significant between-group difference.	Good
Kayondo [19]	2021/Prospective observational study	130	KHQ	All types ≥ 2	Vaginal hysterectomy with sacrospinous ligament vault fixation, vaginal hysterectomy plus anterior and posterior repair with uterosacral ligament vault fixation, posterior colporrhaphy, anterior colporrhaphy	12 months	One year after surgery, the mean QOL scores across all domains and the overall QOL both significantly improved (*p* 0.001). Following surgery, the overall QOL increased by 38.9%.	Good
Favre-Inhofer [36]	2021/Retrospective cohort	59	PDFI-20 + PFIQ	Apical NR	Sacrospinous ligament fixation	60 months	Satisfactory PDFI-20 and PFIQ-7 improvement. No significant difference for the two scores (PFDI-20 and PFIQ-7) between patients with and without recurrence of complications.	Good
Rodrigues [20]	2021/Prospective observational study	129	PISQ-12 + KHQ + P-QoL	All types NR	Tension-free transvaginal tape, inside-out tension-free vaginal transobturator tape, single-incision tape method + vaginal hysterectomy, vaginal colporrhaphy, vaginal vault suspension (sacrocolpopexy)	12 months	PISQ-12 improvement from baseline mean 30.6 (SD 7.3) to 36.1 (SD 5.0); QoL questionnaires (KHQ+ P-QoL) showed a significant improvement (*p* < 0.001)	Good
Belayne [23]	2021/Longitudinal study	224	P-QoL + POP-SS + BIPOP + PHQ-9 + PGI	All types ≥ 2	Conventional anterior-posterior colporrhaphy or uterine preserving sacrospinous fixation	3 and 6 months	Improvement in all domains of P-QoL, and 72% meaningful patient satisfaction after 6 months. Improvements in POP-SS, BIPOP, and PHQ-9 scores were also observed.	Good
Bedel [37]	2022/Prospective observational study	46	PFDI + PFIQ + PISQ	Posterior NR	Midline rectovaginal fascial plication	12 months	Improvement in PFDI-20 and PFIQ-7 scores.	Good

**Table 3 jcm-11-07166-t003:** The characteristics of the studies on the effectiveness of reconstructive vaginal surgeries with mesh on the quality of life in women with pelvic organ prolapse.

First Author [Ref]	Year/Study Design	Sample Size	Questionnire	Prolapse Type and Stage	Intervention	Follow Up	Result	Quality
El Haddad [44]	2012/ Prospective observational study	69	PISQ-12 + PFDI-20	Anterior ≥ 2	Anterior vaginal mesh repair	6 months	Significant decrease in QoL scores and significant increase in sexual activity.	Fair
Yesil [83]	2013/ Prospective observational study	60	German Pelvic Organ Prolapse Questionnaire	Anterior ± Apical ≥ 2	Mesh + anterior repair, anterior repair + posterior repair, posterior repair	12 months	Significant decrease in QoL scores and significant increase in sexual activity.	Fair
Thomin [55]	2013/ Prospective observational study	99	PFDI-20 + PFIQ-7 + PISQ-12	Anterior 2 and 3	Anterior vaginal mesh repair	6 months and then annually	Improvement in PFDI and PFIQ after surgery (*p* < 0.001); PISQ did not change.	Fair
Bartuzi [26]	2013/ Prospective observational study	113	SF-36	All types ≥ 3	Anterior repair + posterior repair + mesh anterior repair + mesh posterior repair + mesh, total repair	6–8 weeks, 16–18 weeks	Improvement in 3 individual domains (general health (GH), vitality (V) and mental health (MH)) and in one summary domain (MCS).	Good
Hefni [21]	2013/ Cross-sectional	127	Euro Qol EQ-5D + SPSQ + KHQ + PSI	Anterior NR	Anterior vaginal mesh repair	6 weeks, 6 and 12 months	Patients reported good current QoL and high patient satisfaction.	Fair
Brocker [78]	2015/ Prospective observational study	69	P-QOL	All types ≥ 2	Vaginal mesh repair	12 months	Significant improvement in all domains of P-QoL.	Good
Husch [32]	2016/ Prospective observational study	148	PGI + P-QOL + ICIQ-VS	Anterior ± Apical All stages	Anterior transvaginal mesh repair	27.2 months	The results of the “prolapse-quality of life” questionnaire were comparable to asymptomatic women.	Good
Buca [69]	2016/ Prospective observational study	116	P-QOL	All types ≥ 2	Anterior repair + mesh, anteroposterior repair + mesh, anteroposterior repair + hysterectomy + mesh	24 months	Improvement in “general state of patients health” (*p* < 0.05); physical, social, and psychological quality of life (*p* < 0.05); significant reduction in the percentage of patients with urinal disturbances (86.2% preoperative vs. 20.7% postoperative; *p* < 0.05); and an improvement in patient’s sexual activity.	Fair
Lamblin [42]	2016/Retrospective, nonrandomized study	195	PFDI-20 + PFIQ-7 + PISQ-12	Anterior + Apical ≥ 3	Anterior and apical repair	12 and 24 months	Function improved, with significantly better PFIQ-7 (*p* = 0.03) and PFDI-20 (*p* = 0.02) scores in the mesh group.	Good
Fünfgeld [68]	2017/Prospective observational study	289	P-QOL	Anterior ≥ 2	Cystocele correction	36 months	Quality of life scores improved significantly in all domains, including sexuality and personal relationships (*p* < 0.001, Wilcoxon test).	Good
Rahkola-Soisalo [43]	2017/ Prospective observational study	207	PFDI-20 + PFIQ-7 + PISQ-12	NR ≥ 2	Transvaginal mesh repair	12 months	Overall postoperative improvement in quality of life (*p* < 0.001).	Fair
Kinjo [12]	2018/Retrospective cohort	237	FSFI + P-QOL	All types ≥ 2	Anterior trans vaginal mesh, anteroposterior trans vaginal mesh, C-trans vaginal mesh, posterior trans vaginal mesh +- mid-urethral sling	12 months	All P-QOL dimensions were significantly improved after surgery. Overall, 79 patients completed the FSFI, and 14 (17.7%) were sexually active. The overall scores for sexual function were significantly improved after surgery.	Fair
Alt [67]	2018/ Prospective observational study	130	P-QOL	Anterior ± posterior ≥ 2	Anterior, posterior, or combined anterior/posterior mesh repair	12 weeks, 12 months, 60 months	Compared to the pre-surgical results, the scores are lowest (QOL increased) at 12 weeks after surgery and slightly increase up to 20–60% of the pre-surgical score results at the five-year follow-up (QOL decreased).	Good
Altman [41]	2018/ Prospective observational study	207	PFDI-20	Apical ± anterior ≥ 2	Uphold™ vaginal support system	12 months	Prolapse-related 15-dimensional instrument measures (excretion, discomfort, sexual activity, distress, and mobility) were significantly improved after surgery (*p* <0.05–0.001). Significant inverse associations were detected between increased 15D scores and a decrease in PFDI-20 and subscale scores (*p* < 0.001), indicating improvements on both instruments.	Good
Cadenbach [7]	2019/ Prospective observational study	54	P-QOL	Anterior ≥ 2	Anterior repair	12 months	An improvement in quality of life could be determined during the study in all domains investigated (*p* < 0.001, Wilcoxon test).	Fair
Sukgen [25]	2020/ Prospective observational study	36	SF-36	NR ≥ 2	Not specified	6–8 weeks- 16–18 months	Some quality-of-life domains (i.e., vitality and mental health), as well as physical and mental health summary scores, improved significantly.	Fair
Zalewski [35]	2020/Before after study	60	P-QOL + SWLS	Apical ≥ 3	Sacrospinous ligament fixation	12 months	Nearly all domains in questionnaires were statistically improved after surgery.	Good
Sukgen [85]	2021/Prospective observational study	72	FSFI	NR NR	Vaginal mesh surgery	3 and 12 months	Positive change in quality of life and sexual function of patients following the intervention. Significant increase according to FSFI score among all endpoints (16%, 86%, and 100% respectively, *p* = 0.001).	Fair
Naumann [82]	2021/Retrospective cohort	107	German Pelvic Organ Prolapse Questionnaire	Anterior ± apical All stages	Anterior repair ± apical in recurrent or complex prolapse	18 months	The total score of the GPOP-Q decreased significantly by 6.4 (*p* < 0.001). Improvement in each domain was observed.	Good
Deltetto [62]	2021/Prospective observational study	15	PISQ-12	Posterior 2	Posterior vaginal repair	12 months	The quality of life was significantly improved in the majority of patients (*p* < 0.05).	Fair
Nakai [29]	2021/ Prospective observational study	28	SF-12	Anterior or apical ≥ 3	Vaginal hysterectomy and utero-sacral ligament colpopexy	12 months	Postoperative QOL was improved in all eight domains after surgery in both synthetic polypropylene and polytetrafluoroethylene mesh groups.	Good
Lo [39]	2021/Retrospective cohort	83	PFDI-20+ PISQ-12	Anterior or apical ≥ 3	Vaginal hysterectomy, mesh repair, posterior colporrhaphy	6 and 12 months	Significant improvement in QoL questionnares.	Good
Chiang [40]	2021/Randomized controlled trial	64	PFDI-20 + PFIQ-7	Anterior ≥ 2	Transvaginal mesh with concomitant midline fascial plication	6, and 12 months	Both groups were improved regarding QoL.	Fair
Lucot [9]	2022/Randomized controlled trial	262	PFDI + PFIQ + FSFI + PISQ-IR + EQ-5D-3L + ICIQ-UI	Anterior ≥ 2	Transvaginal mesh surgery vs. laparoscopic mesh sacropexy	18, 24, 36, and 48 months	Laparoscopic sacropexy acted better than trans-vaginal mesh surgery.	Good

**Table 4 jcm-11-07166-t004:** The characteristics of the studies on the effectiveness of obliterative vaginal surgeries on the quality of life in women with pelvic organ prolapse.

First Author [Ref]	Year/Study Design	Sample Size	Questionnaire	Prolapse Type and Stage	Intervention	Follow Up	Main Findings	Quality
Yeniel [73]	2012/Prospective observational study	10	P-QOL	NR ≥ 3	Le Fort colpocleisis	6 months	Reduced scores of P-QoL, reflecting improvement in QoL following colpocleisis.	Good
Katsara [86]	2016/Retrospective study	44	Author’s design	All types NR	Le Fort colpocleisis	13 years	Overall, 75% reported a positive impact on QoL, 10% a negative because of urinary problems, 10% could not report any change in the QoL, and 5% could not answer this question.	Good
Petcharopas [72]	2018/Retrospective cohort study	295	P- QOL	NR ≥ 1	Le Fort colpocleisis vs. vaginal hysterectomy, sling, anterior repair, posterior repair, McCall culdoplasty, vaginal vault suspension	23.87 months	P-QOL scale revealed significantly less postoperative impairment in the obliterative surgery group (1.75 vs. 5.26, *p* = 0.023). There were no significant differences in other P-QOL domains. Le Fort acted better in elderly.	Good
Ertas [71]	2021/Retrospective cohort study	53	P-QOL	All types ≥ 3	Le Fort colpocleisis	1, 6, and 12 months	Significant improve in the postoperative P-QoL score (*p* < 0.001).	Fair
Karaca [79]	2021/Retrospective cohort study	98	P-QOL	NR NR	Le Fort colpocleisis vs. sacrospinus fixation	Immediately after surgery	Le Fort acted better in elderly	Fair
Farghali [18]	2021/Prospective comparative study	86	WHOQOL- BREF	All types ≥ 2	Le Fort colpocleisis vs. sacrospinous fixation	6 weeks; 3, 6, 12 months	Sacrospinous fixation acted better than Le Fort. Psychological and social health domains were significantly higher in the sacrospinous fixation group compared to the Le Fort group (p2 = 0.04 and 0.02, respectively). General health satisfaction and total QoL scores were significantly higher in the sacrospinous fixation group compared to the Le Fort group (p2 = 0.03 and 0.01, respectively).	Fair
Agacayak [27]	2022/Prospective comparative study	46	SF-36 + UDI + PFIQ	Anterior ± Posterior ≥ 3	Le Fort colpocleisis vs. sacrospinus fixation	6 months, 2 and 5 years	Le Fort acted better than sacrospinus fixation. Pelvic Floor Impact Questionnaire and SF36 were found to be significantly better in the Le Fort group (*p.* = 0.039 and 0.042, respectively).	

**Table 5 jcm-11-07166-t005:** The characteristics of the studies on the effectiveness of abdominal surgeries on the quality of life in women with pelvic organ prolapse.

First Author [Ref]	Year/Study Design	Sample Size	Questionnaire	Prolapse Type and Stage	Intervention	Follow Up	Main Findings	Quality
Thibault [56]	2013/Prospective observational cohort	148	PISQ-12+ PFDI-20+ PFIQ-7	All types ≥ 2	Laparoscopic sacrocolpopexy	12 months	Improvement in PISQ-12, PFDI-20, and PFIQ-7 total mean scores. (*p* < 0.05).	Good
Majkusiak [88]	2015/Observational study	40	10-point analogue scale-sex-QOL	Apical ≥ 3c	Abdominal cervicosacropexy (laparotomy)	18 months	The average score of the quality of sexual life was 5.75 (SD 2.52, 95% CI: 4.41–7.1) before and increased to 7.93 (SD 1.77, 95% CI: 6.9–8.95) after the procedure (*p* < 0.05). The mean score of the overall QoL in relation to POP before and after the surgery was 2.77 (SD 2.39, 95% CI: 1.87–8.64) and 9.03 (SD 1.08, 95% CI: 8.66–9.43), respectively (*p* < 0.001).	Good
Linder [54]	2015/Retrospective observational cohort	84	PFIQ-7 + PFDI-20	Apical ≥ 3	Robotic sacrohysteropexy	72 months	Minimal postoperative pelvic floor complications and high rate of patient satisfaction achieved.	Good
Grimminack [53]	2016/Prospective observational cohort	50	PFDI-20	Anterior ± posterior > 2	Robotic sacrohysteropexy	60 months	Improvement in the QoL following surgery (*p* < 0.05).	Good
Nguyen [52]	2018/Prospective observational cohort	222	PFDI	All types ≥ 1	Vaginal and abdominal repair	12 months	Improvement in QoL of both groups after 1 year. Vaginal group showed higher QoL scores (45.6 vs. 32.6, *p* = 0.032).	Fair
Tahaoglu [76]	2018/Observational study	22	FSFI + P-QOL	All types ≥ 2	Laparoscopic pectopexy	6 months	FSFI and P-QOL scores improved significantly following surgery (*p* < 0.05).	Fair
Pizzoferrato [33]	2019/Observational study	152	PFDI-20 + ICIQ-UI SF + PFIQ-7 + Euro Qol EQ-5D + PISQ-12 + PGI-1	All types ≥ 2	laparoscopic sacrocolpopexy	48 months	Improvement in PFDI-20 (median: 47.4 before surgery vs. 34.4 afterwards, *p* = 0.002); improvement in PGI-I score = 1.8 ± 1.1.	Fair
MäkeläKaikkonen [51]	2019/Prospective randomized study	30	PFDI-20+ PFIQ-7	Posterior ≥ 2	Laparoscopic and robotic ventral mesh rectopexy	24 months	Both were comparable	Good
Luque [87]	2020/Prospective observational cohort	33	Pac-QoL	Apical + posterior ≥ 3	Laparoscopic sacrocolpoperineopexy	6, 12, and 36 months	Significant improvement in QoL scores at 6, 12, and 36 months following surgery.	Fair
Zhu [45]	2021/Retrospective study	50	PFIQ-7 + PFDI-20	Apical ≥ 2	Modified laparoscopic uterine suspension and vaginal hysterectomy and sacrospinous ligament fixation	12 months	Improvement in QoL (P < 0.001 for PFIQ-7 and PFDI-20).	Fair
Van Zanten [46]	2021/Prospective observational cohort	77	PFDI-20 + PFIQ-7	Apical ≥ 2	Robot-assisted sacrocolpopexy and supracervical hysterectomy with sacrocervicopexy	50 months	Improvement in median UDI-6 scores (26.7 vs. 22.2, *p* = 0.048), and median PFIQ-7 scores (60.0 vs. 0, *p* = 0.008).	Good
Sliwa [47]	2021/Observational cohort	42	P-QOL + PFDI-20	All types ≥ 3	Laparoscopic anterior fixation with mesh	12 months	Improvement in P-QoL score, as well as PFDI-20, along with its 3 symptom scales.	Fair
Karsli [48]	2021/Observational cohort	31	P-QOL + PISQ-12 + PFDI-20	Apical ≥ 3	Laparoscopic pectopexy	3 months	Improvement in P-QOL (from 83.45 ± 8.7 (64–98) to 8.61 ± 6.4 (0–23) (*p* < 0.05)); in PISQ-12 from 29.61 ± 4.8 (14–38) to 7.1 ± 3.2 (1–13)). Significant improvement in UDI-6 after procedure.	Fair
Cengiz [49]	2021/Randomized controlled Study	49	ICIQ-VS+ ICIQ-QOL+ PFDI-20	Apical ≥ 2	Vaginal assisted laparoscopic sacrohysteropexy vs. vaginal hysterectomy with vaginal vault suspension	12 months	ICIQ-VS score, ICIQ-QOL, UDI-6, and IIQ-7 scores were improved for both groups.	Good
Akbaba [74]	2021/Retrospective study	37	P-QOL	Anterior + apical ≥ 3	Laparoscopic lateral suspension with mesh	20 months	Improvement.	Good
Okcu [50]	2021/Prospective observational cohort	65	PISQ-12+ PFDI-20	All types Any stages	Vaginal hysterectomy with sacrospinous ligament fixation, laparoscopic hysterectomy with sacrocolpopexy or abdominal hysterectomy with sacrocolpopexy	36 months	Abdominal group acted better (*p* = 0.047), while the sexual function was similar.	Good
Obut [75]	2021/ Prospective randomized study	62	FSFI + P-QOL	Apical ≥ 2	Laparoscopic pectopexy-hysterosacropexy	12 months	Improvement in all domains of POP-Q, P-QOL, and FSFI scores in both groups.	Good
D’altilia [81]	2022/Prospective randomized study	95	PFIQ-7	Anterior + Apical NR	Laparoscopic, robotic or abdominal sacrocolpopexy	36 months	Single anterior vaginal mesh acted better than anterior/posterior mesh.	Good

**Table 6 jcm-11-07166-t006:** The characteristics of the studies on the effectiveness of pessary on the quality of life in women with pelvic organ prolapse.

First Author [Ref]	Year/Study Design	Sample Size	Questionnaire	Prolapse Type and Stage	Intervention	Follow Up	Result	Quality
Manchana [77]	2012/Retrospective observational study	126	P-QOL	NR ≥ 2	Ring pessary	12 months	The median score from all PQOL domains except personal relationships were significantly decreased after 1 year of ring pessary use. The median total scores at baseline and at 1 year were 40 and 8, respectively (*p* < 0.001).	Good
Lone [80]	2015/Prospective observational study	287	ICIQ-VS + ICIQ-UI SF	NR ≥ 1	Pessary (ring/cub/Gellhorn) vs. surgery	12 months	Both groups were improved regarding QoL. No statistically significant between-group difference was noted.	Good
Tenfelde [59]	2015/Observational cross-sectional study	56	PFDI-20 + PFIQ-7	NR NR	Pessary	-	Overall, women reported high levels of QOL. Health-related QoL scores correlated negatively with PFDI-20 scores (ρ = −0.55, *p* = 0.001). Women with higher QoL reported fewer pelvic floor symptoms for each of the subscales (urinary, colorectal, and prolapse).	Fair
Coelho [28]	2018/Prospective observational study	19	ICIQ-VS + SF-36	Anterior±apical ≥ 3	Ring pessary	6 months	SF-36 had significant improvement in three specific domains: general state of health (*p* = 0.090), vitality (*p* = 0.0497), and social aspects (*p* = 0.007). ICIQ-VS presented a reduction in the vaginal symptoms (*p* < 0.0001) and an improvement in QoL (*p* < 0.0001).	Fair
Yang [58]	2018/Retrospective study	300	PFDI-20 + PFIQ-7	All types NR	Ring—Gellhorn	6 months	Improvement.	Good
Mao [89]	2018/Prospective observational study	142	PFDI-20 + PFIQ-7	Anterior Posterior ≥ 3	Ring with support pessary, follow-up with Gellhorn if failed	13–24 months	Improvement in both questionnaires. All domains were improved significantly (*p* < 0.001).	Good
Bradley [57]	2021/Cross-sectional analysis	568	PFDI-20 + PFIQ-7 + PISQ-12 + ICIQ-UI-SF	All types Any stage	Pessary vs. surgery (both reconstructive and obliterative)	36 months	Women choosing POP surgery vs. pessary had similar physical and mental generic QOL.	Good
Mendes [22]	2021/Prospective observational study	50	SF-36 + P-QOL + KHQ	All types Any stages	Ring pessary	4 months	Improvement in the general and specific quality of life of women.	Fair
Thys [34]	2021/Prospective observational study	291	PGI-I + DSQOL	All types ≥ 2	Ring pessary with or without knob-space filling pessary	12 months	Improvement.	Good
Carlin [84]	2021/Retrospective study	130	German Pelvic Organ Prolapse Questionnaire	All types Any stages	Pessary vs. surgery	3 months	Surgery acted better than pessary.	Fair
Zeiger [30]	2022/Multicenter, longitudinal, prospective observational study	97	ICIQ-VS + SF-12	NR ≥ 3	Ring pessary	6 months	Improved sexual function (78.6%), vaginal symptoms (91.8%), and quality of life (92.8%) (*p* < 0.01) after the intervention.	Good

## Data Availability

Data is available due to request.

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
