# Peer review of "Quality of Life Following Pelvic Organ Prolapse Treatments in Women: A Systematic Review and Meta-Analysis"

_jcm, 2022, doi:10.3390/jcm11237166_

Round 1
Reviewer 1 Report
Manuscript titled " Quality of life in women with pelvic organ prolapse: A systematic review and meta-analysis" describe in a satisfactory manner the quality of life of woman with pelvic prolapse. The overall structure is of good quality, methods and conclusion are clear to readers, however, some changes are needed:
1. a description of cardiovascular diseases in these patients should be made,
2. a description of quality of life and cardiovascular complications in cancer patients in urology department should be made and a possible correlation with immunotherapy-related side effects ( cite doi: 10.3390/jpm10040179. )
3. a description of diabetes-related cardiovascular complications should be also done. specifically, authors should describe the role of hyperglycemia in quality of life in these patients and how new antidiabetic drugs like gliflozins should improve quality of life and cardiovascular complications ( cite doi: 10.1186/s12933-021-01346-y.)
Author Response
#Reviewer 1
Manuscript titled " Quality of life in women with pelvic organ prolapse: A systematic review and meta-analysis" describe in a satisfactory manner the quality of life of woman with pelvic prolapse. The overall structure is of good quality, methods and conclusion are clear to readers, however, some changes are needed:
- a description of cardiovascular diseases in these patients should be made,
Response: Thank you for this comment. I believe that some type of non-surgical modalities such as pessary is the best choice for older women that most of them have some type of cardiovascular diseases.
- a description of quality of life and cardiovascular complications in cancer patients in urology department should be made and a possible correlation with immunotherapy-related side effects ( cite doi: 10.3390/jpm10040179. )
Response: Thank you, I added this citation in discussion part.
- a description of diabetes-related cardiovascular complications should be also done. Specifically, authors should describe the role of hyperglycemia in quality of life in these patients and how new antidiabetic drugs like gliflozins should improve quality of life and cardiovascular complications ( cite doi: 10.1186/s12933-021-01346-y.)
Response: Thank you, I added this citation in discussion part.
Reviewer 2 Report
TITLE AND ABSTRACT
-Try to include in the title that you assess the quality of life associated with prolapse treatments
-In the summary, please consider whether to change the expression “the use for this technique” to “the use of this instrument/device”
-76 may be too large a number to include in a review. Assess whether the objective of the review and the inclusion criteria of the study are too broad/inclusive
-Consider whether the questionnaires studied actually assess quality of life or symptom impact (or at least quality of life related to symptom impact)
-Consider whether the last sentence of the conclusion is derived directly from your study
INTRODUCTION
-Check if reference 3 is adequate to provide prevalence data
-Why do you use capital letters the first time they mention pessaries?
-The sentence “with a long antiquity since the beginning of recorded history” may be misunderstood, try to rewrite it
-Are POPDI and POPIQ questionnaires or dimensions of other questionnaires? There is no concordance with what you say in the summary. I may be wrong
-Try to explain briefly what are the usual surgeries for pop and at least what is a pessary
-Nothing is mentioned about conservative treatment by physiotherapy/exercise
METHODS
-You no longer use the usual abbreviations
-Consider whether it is necessary to specify which designs you included in their review, not just which ones you excluded
-The sentence “the references of the selected articles also were examined for possible publications manually” is not well understood
-Shouldn't you have included in your eligibility criteria that the studies have used questionnaires to assess quality of life? Understood but not specified
-Consider whether to change “The initial search was undertaken in 3 main databases including PubMed, Scopus and Web of Science from 1st January 2012 until 31th June 2022” to “The initial search was undertaken in 3 main databases including articles published in PubMed, Scopus and Web of Science from 1st January 2012 until 31th June 2022”
-Please specify the search strategy(ies) used in each database so that the review can be replicated
-Explain a little more about the NIH tool. Check if reference 14 is valid here
RESULTS
-In figure 1 you have a blue mark that should be removed from the image. What does RPL mean in the figure. This flow chart specifies exclusion criteria that were not clearly specified in the methods section. It is important that you correct this
-In figure 1 it does not make sense that below they specify causes for exclusion when the number of excluded is zero, but above you reject 30 and 27 and in the figure you do not specify the reasons. What do the asterisks mean?
-Figure 2: obliterative also in capital letters
-Table 2-6: in the "Result" column you can be more specific, giving quantitative values of the improvement. Can you explain what the abbreviation NR stands for? It is necessary to specify, either in the tables or in the text, the design of the items included.
DISCUSSION
-Check if there are any related reviews that you can cite in your discussion.
Author Response
#Reviewer 2
TITLE AND ABSTRACT
Try to include in the title that you assess the quality of life associated with prolapse treatments
Response: I corrected it. Quality of life following pelvic organ prolapse treatments in women: A systematic review and meta-analysis
In the summary, please consider whether to change the expression “the use for this technique” to “the use of this instrument/device”
Response: thank you, I did it.
-76 may be too large a number to include in a review. Assess whether the objective of the review and the inclusion criteria of the study are too broad/inclusive
Response: This is exactly true. But we decided to cover all new treatment and because this is hot topic the number of the articles were high. We tried to explore very carefully.
-Consider whether the questionnaires studied actually assess quality of life or symptom impact (or at least quality of life related to symptom impact)
Response: Thank you for this comment. All questionnaires assess the quality of life in general or some specific aspect of quality of life like urinary or sexual assessment.
-Consider whether the last sentence of the conclusion is derived directly from your study
Response: Thank you. This section was rewritten.
INTRODUCTION
-Check if reference 3 is adequate to provide prevalence data
Response: Thank you, I deleted the sentence.
-Why do you use capital letters the first time they mention pessaries?
Response: I appreciate and corrected it.
-The sentence “with a long antiquity since the beginning of recorded history” may be misunderstood, try to rewrite it
Response: Thank you, I revised it.
-Are POPDI and POPIQ questionnaires or dimensions of other questionnaires? There is no concordance with what you say in the summary. I may be wrong.
Response: Pelvic Organ Prolapse Distress Inventory 6 (POPDI-6) and Pelvic Organ Prolapse Impact Questionnaire (POPIQ-7) are dimensions of the Pelvic Floor Distress Inventory (PFDI-20) and Pelvic Floor Impact Questionnaire (PFIQ-7) questionnaires, respectively. The POPDI-6 and POPIQ-7 are specifically designed for POP (1).
-Try to explain briefly what are the usual surgeries for pop and at least what is a pessary
Response: I added them in the introduction.
-Nothing is mentioned about conservative treatment by physiotherapy/exercise
Response: Thank you, they are added either.
METHODS
-You no longer use the usual abbreviations
Response: Thank you, I revised the whole manuscript to use the abbreviations wherever needed.
-Consider whether it is necessary to specify which designs you included in their review, not just which ones you excluded
Response: Thank you, the whole section was rewritten.
-The sentence “the references of the selected articles also were examined for possible publications manually” is not well understood
Response: Thank you, by that sentence we meant we also checked the reference section of related studies in order to include possible studies which has not been identified through our electronic search.
-Shouldn't you have included in your eligibility criteria that the studies have used questionnaires to assess quality of life? Understood but not specified
Response: Thank you, I specified in the relevant section.
-Consider whether to change “The initial search was undertaken in 3 main databases including PubMed, Scopus and Web of Science from 1st January 2012 until 31th June 2022” to “The initial search was undertaken in 3 main databases including articles published in PubMed, Scopus and Web of Science from 1st January 2012 until 31th June 2022”
Response: Thank you, changes amended.
-Please specify the search strategy(ies) used in each database so that the review can be replicated
Response: Full search strategy is provided in supplementary material.
-Explain a little more about the NIH tool. Check if reference 14 is valid here
Response: Thank you, The risk of bias was assessed using National, Heart, Lung, and Blood Institute (NHLBI) tools for cohort and cross-sectional studies (2). NHLBI risk of bias tools contains 14 signaling questions for the assessment of the quality of included studies.
RESULTS
-In figure 1 you have a blue mark that should be removed from the image. What does RPL mean in the figure. This flow chart specifies exclusion criteria that were not clearly specified in the methods section. It is important that you correct this
Response: Thank you, this was a critical error from the author’s side. Figure 1 redesigned with its true values.
-In figure 1 it does not make sense that below they specify causes for exclusion when the number of excluded is zero, but above you reject 30 and 27 and in the figure you do not specify the reasons. What do the asterisks mean?
Response: Figure 1 redesigned with its true values.
-Figure 2: obliterative also in capital letters
Response: Thank you, changes amended.
-Table 2-6: in the "Result" column you can be more specific, giving quantitative values of the improvement. Can you explain what the abbreviation NR stands for? It is necessary to specify, either in the tables or in the text, the design of the items included.
Response: Thank you for your insightful comment. We added a column named “Main findings” to report the final result of each study. Furthermore, we tried to pool the quantitative results of the most used questionnaires (PFDI-20 and PFIQ-7) in our meta-analysis. It is worth mentioning, reporting quantitative value of other questionnaires could not be done because most of them had data on several dimensions without a total score.
NR stands for not reported and we provided abbreviation meanings for tables.
We added the study design for each study in “Year/Study design” column.
DISCUSSION
-Check if there are any related reviews that you can cite in your discussion.
Response: Thank you, following reviews were cited:
-Raju R, Linder BJ. Evaluation and Management of Pelvic Organ Prolapse. Mayo Clin Proc. 2021 Dec;96(12):3122-3129. doi: 10.1016/j.mayocp.2021.09.005. PMID: 34863399.
-Verbeek M, Hayward L. Pelvic Floor Dysfunction And Its Effect On Quality Of Sexual Life. Sex Med Rev. 2019 Oct;7(4):559-564. doi: 10.1016/j.sxmr.2019.05.007. Epub 2019 Jul 24. PMID: 31351916.
-Mowat A, Maher D, Baessler K, Christmann-Schmid C, Haya N, Maher C. Surgery for women with posterior compartment prolapse. Cochrane Database Syst Rev. 2018 Mar 5;3(3):CD012975. doi: 10.1002/14651858.CD012975. PMID: 29502352; PMCID: PMC6494287.
- Barber MD, Walters MD, Bump RC. Short forms of two condition-specific quality-of-life questionnaires for women with pelvic floor disorders (PFDI-20 and PFIQ-7). Am J Obstet Gynecol. 2005;193(1):103-13.
- National Heart L, Institute B. Quality assessment tool for observational cohort and cross-sectional studies [Internet]. Bethesda (MD): National Heart, Lung, and Blood Institute (NHLBI) [Available from: https://www.nhlbi.nih.gov/health-topics/study-quality-assessment-tools.